METHODS

# HAPP: High-accuracy pipeline for processing deep metabarcoding data

John Sundh[1☉], Emma Granqvist[2☉], Ela Iwaszkiewicz-Eggebrecht[2], Lokeshwaran Manoharan[3], Laura J. A. van Dijk[2], Robert Goodsell[4], Nerivania N. Godeiro[5], Bruno C. Bellini[6], Johanna Orsholm[4], Piotr Łukasik[7], Andreia Miraldo[2,8], Tomas Roslin[4], Ayco J. M. Tack[9], Anders F. Andersson[10☉*], Fredrik Ronquist[2☉*]

**1** Department of Biochemistry and Biophysics, National Bioinformatics Infrastructure Sweden, Science for Life Laboratory, Stockholm University, Solna, Sweden, **2** Department of Bioinformatics and Genetics, Swedish Museum of Natural History, Stockholm, Sweden, **3** Department of Laboratory Medicine, National Bioinformatics Infrastructure Sweden, Science for Life Laboratory, Lund University, Lund, Sweden, **4** Department of Ecology, Swedish University of Agricultural Sciences (SLU), Uppsala, Sweden, **5** Shanghai Natural History Museum, Shanghai Science & Technology Museum, Shanghai, People's Republic of China, **6** Department of Botany and Zoology, Biosciences Center, Federal University of Rio Grande do Norte, Natal, Brazil, **7** Institute of Environmental Sciences, Faculty of Biology, Jagiellonian University, Krakow, Poland, **8** Biodiversity and Sustainability Solutions, Aveiras de Baixo, Portugal, **9** Department of Ecology, Environment and Plant Sciences, Stockholm University, Stockholm, Sweden, **10** Department of Gene Technology, Science for Life Laboratory, KTH Royal Institute of Technology, Stockholm, Sweden

☉ These authors contributed equally to this work.
* fredrik.ronquist@nrm.se (FR); anders.andersson@scilifelab.se (AFA)

## Abstract

Deep metabarcoding offers an efficient and reproducible approach to biodiversity monitoring, but noisy data and incomplete reference databases challenge accurate diversity estimation and taxonomic annotation. Here, we introduce a novel algorithm, NEEAT, for removing spurious operational taxonomic units (OTUs) originating from nuclear-embedded mitochondrial DNA sequences (NUMTs) or sequencing errors. It integrates 'echo' signals across samples with the identification of unusual evolutionary patterns among similar DNA sequences. We also extensively benchmark current tools for chimera removal, taxonomic annotation and OTU clustering of deep metabarcoding data. The best performing tools/parameter settings are integrated into HAPP, a high-accuracy pipeline for processing deep metabarcoding data. Tests using CO1 data from BOLD and large-scale metabarcoding data on insects demonstrate that HAPP significantly outperforms existing methods, while enabling efficient analysis of extensive datasets by parallelizing computations across taxonomic groups.

**Data availability statement:** Open source code is available at: https://github.com/insect-biome-atlas/happ.

**Funding:** This work was supported by the SciLifeLab & Wallenberg Data Driven Life Science Program, the Knut and Alice Wallenberg Foundation (grants KAW 2020.0239 and KAW 2017.0003 to FR), the Swedish Research Council (grants 2018-04620 and 2021-04830 to FR, grant 2021-05563 to AFA), the National Bioinformatics Infrastructure Sweden (NBIS) at SciLifeLab, and the Finance to Revive Biodiversity (FinBio) program, financed by Mistra – the Swedish Foundation for Strategic Environmental Research (DIA 2020/10 to FR). The funders had no role in study design, data collection and analysis, decision to publish, or preparation of the manuscript.

**Competing interests:** The authors have declared that no competing interests exist.

## Author summary

Charting and monitoring biodiversity is essential for understanding and protecting ecosystems, but it has been difficult to collect data cost-efficiently at scale. An approach that potentially solves this problem is metabarcoding—a method that can be applied to DNA from environmental samples to identify many species at once. Unfortunately, it may produce misleading results due to noise in the data. A particularly challenging problem when analysing data from mitochondrial DNA, such as the CO1 gene often used for analysing insect biodiversity, is the existence of nuclear encoded copies of the gene that can severely inflate diversity estimates. We created an algorithm called NEEAT that helps remove such misleading signals by combining information from multiple samples and spotting unusual patterns of genetic change. We also tested many existing tools for other steps of data processing, and combined NEEAT with the best tools in creating a new, high-accuracy analysis pipeline we call HAPP. Using both simulated and real-world insect data, we show that our approach is not only more accurate than current methods but also efficient at handling large datasets. Our work aims to make biodiversity studies more precise and scalable, supporting better conservation and environmental decision-making.

## Introduction

DNA metabarcoding—PCR amplification of a marker gene from an environmental sample followed by high-throughput sequencing [1]—has revolutionized the collection of biodiversity data thanks to its efficiency and cost-effectiveness. However, the data can be noisy [2–4], even after denoising it into amplicon sequence variants (ASVs). In addition to authentic ASVs, ASVs representing chimeras, PCR and sequencing errors, and off-target sequences such as NUMTs [5], are typically present. Interpreting the authentic ASV data presents additional challenges. Identifying species lacking reference sequences typically involves *de novo* clustering of ASVs into operational taxonomic units (OTUs), requiring optimized algorithms to ensure accurate species-OTU correspondence. Finally, the short sequence lengths and incomplete reference libraries complicate taxonomic annotation.

We are now seeing a significant expansion in the scope of DNA metabarcoding projects—the biomes investigated are sampled more intensely and samples sequenced more deeply [6,7]. We refer to this as 'deep metabarcoding'. The methods are also increasingly applied to macrobial diversity and appropriate genetic markers, such as the mitochondrial gene cytochrome *c* oxidase 1 (CO1). The advent of deep metabarcoding and its application to macrobial taxa justifies a reassessment of available tools as well as developing new ones.

In the present paper, we present a new tool - NEEAT - for filtering noise OTUs, perhaps the most challenging aspect of processing deep metabarcoding data. We also benchmark a suite of tools for chimera removal, taxonomic annotation and OTU

clustering. The best performing tools are combined with NEEAT in a new high-accuracy pipeline for processing deep metabarcoding data, HAPP.

Off-target sequences are a major source of noise in metabarcoding, and may constitute a larger fraction of unique sequences in deep metabarcoding because of the sequencing depth. For mitochondrial markers, these often include NUMTs—nuclear DNA of mitochondrial origin—that may be present in large numbers in the genome [8]. Over time, NUMTs accumulate mutations, insertions, and deletions, as they are no longer subject to the same selective pressures as functional mitochondrial genes. Typically, NUMTs are less abundant than their mitochondrial counterparts, as the nuclear genome has 100–10,000 [9,10] times fewer copies per cell, but some NUMTs may be present in multiple copies within a genome, affecting the copy ratio [8]. Initially, NUMTs are identical to their mitochondrial origin and pose no issue. With time, they accumulate enough mutations to become distinguishable by frame shifts or premature stop codons [11–14]. The challenge arises in intermediate stages when NUMTs still resemble authentic mitochondrial sequences but differ just enough to be mistaken for distinct species.

NUMTs can be identified using evolutionary signatures, as they differ from functional genes under strong constraining selection. These signatures also help detect amplification and sequencing errors. One method involves fitting a profile hidden Markov model (HMM) to the protein coded by the target gene to flag outliers as potential NUMTs [12]. Another approach, explored here, compares evolutionary and edit distances (number of changes needed to go from one sequence to another) between pairs of similar ASVs, as authentic ASVs should have fewer non-synonymous substitutions relative to the edit distance compared to pairs involving NUMT(s). Alternatively, NUMTs can be detected through distributional patterns, as they should co-occur with their parent sequences but at lower copy numbers. LULU [13] uses this "echo" signal, sequences matching the distribution across samples of more abundant ones that are within a genetic distance cutoff. A simpler method is applying abundance thresholds, as NUMTs are unlikely to generate high-read ASVs [4]. The metaMATE tool [14] optimizes abundance filters based on organism groups and datasets.

Here, we introduce a new noise filtering algorithm, NEEAT (noise reduction using echos, evolutionary signals and abundance thresholds), which combines these techniques, and we benchmark it against LULU, HMM filtering and abundance filtering.

We also reassessed the remaining data processing steps before wrapping NEEAT into a complete data processing pipeline, HAPP. Hybrid ASVs, or chimeras, can form during PCR amplification and be mistaken for novel biodiversity. While standard pipelines include chimera removal, our initial benchmarking showed DADA2's default method was insufficient, suggesting that chimeras may be particularly problematic in deep metabarcoding. To address this, we incorporated an additional chimera removal step in HAPP. Several powerful tools exist, including ChimeraChecker [15], ChimeraSlayer [16] and Perseus [17]. We focused on UCHIME for its accuracy and speed [18]. It detects chimeras by comparing ASVs in descending abundance and scoring alignments between queries and candidate parents. We tested multiple strategies to optimize UCHIME's application to deep metabarcoding data.

Taxonomic annotation in metabarcoding relies on either matching ASVs to a reference library or placing them in a phylogenetic tree. The former is effective when reference libraries are fairly complete, while the latter is more reliable otherwise [19]. For reference library matching, we benchmarked $k$-mer-based and alignment-based methods. For $k$-mer approaches, we tested the DADA2 [20] implementation of the RDP classifier [21], a naive Bayesian $k$-mer approach combined with bootstrapping to estimate confidence, and SINTAX [22] implemented in VSEARCH [23], which applies a non-Bayesian algorithm and bootstrapping. We also evaluated a multinomial naive Bayes classifier implemented in QIIME2 [24]. For alignment-based matching, we focused on VSEARCH through the QIIME2 interface, which uses a BLAST-like alignment approach and computes a consensus taxonomy for the matches. For phylogenetic annotation, we used EPA-NG [19,25] with Gappa [26] which places query sequences in a reference tree with maximum likelihood.

Clustering ASVs into species is another important step in processing metabarcoding data. For CO1 barcode sequences, the refined single linkage (RESL) algorithm has been particularly influential [27]. It clusters CO1 barcode

sequences in the Barcode of Life Data System (BOLD) reference library [28] into putative species, called BINs (Barcode Index Numbers). Deep metabarcoding differs from traditional BOLD data in that it provides a much higher number of sequences per species, which could simplify species recognition and favor alternative clustering approaches. Furthermore RESL is not open-source. We evaluated three open-source clustering algorithms for inclusion in HAPP: SWARM [29,30], OptiClust [31], and dbOTU3 [32,33]. SWARM, which shares many features with RESL, employs an agglomerative single-linkage clustering approach in two phases: an initial growth phase where ASVs are grouped based on sequence similarity, and a breaking phase that refines clusters using abundance data. OptiClust aims to minimize the discord between the desired maximum intra-cluster distance between sequence pairs and the actual pairwise distances within and between clusters, balancing false positives and negatives to optimize clustering [31]. dbOTU3 is unique in incorporating ASV distribution across samples in addition to genetic distance [32,33], an approach that should be particularly well suited for deep metabarcoding datasets.

To properly benchmark any pipeline for deep metabarcoding data, realistic test data is needed, ideally covering a range of taxa and different biogeographic regions. For benchmarking existing and new tools for integration in HAPP, we utilized a large data set recently generated in the Insect Biome Atlas (IBA) project, covering arthropod communities from Sweden and Madagascar [6]. The core data consists of 6,483 weekly Malaise trap samples (4,560 from Sweden and 1,923 from Madagascar)—estimated to contain 9 million specimens and tens of thousands of species—subjected to deep metabarcoding (ca. 1 M read-pairs per sample) of a 418 bp stretch of CO1. Importantly, the data cover a range of diverse taxa with widely different life histories, and two regions with dramatically different coverage in available reference databases. The power of the benchmarking is increased by the fact that the Swedish insect fauna is well characterized, with the inventory of some taxa being virtually complete [34], and that a carefully curated and extensive CO1 reference library is available from the neighbouring country Finland [35]. The benchmarking was further complemented using the entire set of CO1 data from BOLD.

The design of HAPP is based on the results of these efforts. It integrates all tools required for post-processing metabarcoding data into a highly configurable and scalable workflow.

## Results

The HAPP pipeline reconstructs species-level OTUs from deep metabarcoding data. As input, it expects denoised ASVs and their counts across samples from a standard workflow, such as DADA2 [20]. After an initial quality filtering that removes ASVs outside the expected size-range or with stop codons in the expected reading frame, post-processing is applied in the four steps described above: chimera removal, taxonomic annotation, OTU clustering, and noise filtering. We present the benchmarking results for each step first, and then discuss the design of HAPP.

### Extra chimera filtration for improved data quality

We benchmarked four different strategies for removing chimeras from denoised ASV data. Specifically, we assessed whether chimera removal could be improved in deep metabarcoding through more reliable detection of chimeric parents across all samples (batchwise method), compared to relying entirely on finding parents in the same sample (samplewise method). For both methods, we tested lenient and strict requirements on chimera-parent co-occurrence (see Methods). We found that the two samplewise methods removed considerably more ASVs than the batchwise methods, while simultaneously improving the clustering results as indicated by the higher recall and smaller cluster:species ratios (both metrics reflecting the degree to which OTUs are matching and not over-splitting species; Fig 1). Thus, the samplewise methods had a better signal to noise ratio. The strict samplewise method performed slightly better than the lenient alternative in terms of trusted ASVs (see Methods) or reads removed. Enforcing a strict chimera-parent matching criterion improved the false positive ratio considerably for the batchwise method (reflected by the smaller number of trusted ASVs and reads removed) but it did not improve the false negative ratio (reflected by the number of ASVs and proportion abundant

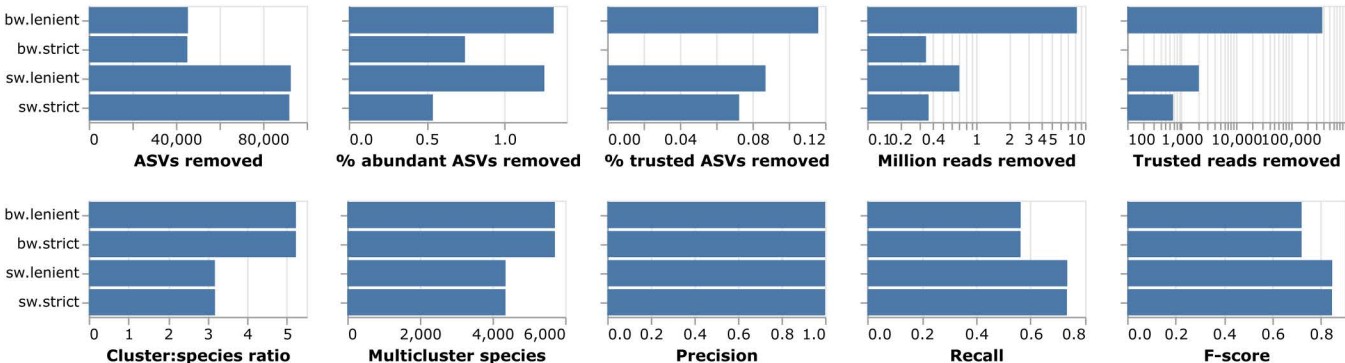

**Fig 1. Benchmarking of chimera removal methods.** Each panel shows a quality metric for the four chimera removal methods tested (see Methods for detailed description of the metrics). The benchmarking was run on the early version of the Swedish IBA data (636,297 ASVs).

ASVs removed), or the clustering results. In conclusion, the strict samplewise method performed best; it removed a large amount of chimeric sequences while retaining the vast majority of authentic ASVs.

### Improved taxonomic classification by combining *k*-mer and phylogenetic approaches

To benchmark the different taxonomic classification algorithms, we used a customized version of the BOLD database (see Methods) and generated different query sequences/reference database combinations, each with 1000 randomly sampled query sequences with a known taxonomy obtained from BOLD. These ranged from the easiest (case 1), where identical sequences to the queries remained in the database, to the hardest (case 5), where sequences of the same orders but not families remained in the database (Fig 2a). For both query and database sequences, only the 418 bases of the CO1 gene corresponding to the barcode region sequenced in the IBA project were used.

We found that the *k*-mer based tools SINTAX [22] ('vsearch_sintax') and SKLEARN ('qiime2_sklearn') had the highest number of correctly assigned species when all sequences were kept in the reference database (case 1) and when sequences identical to the query sequence were removed but sequences belonging to the same species remained in the database (case 2; Fig 2a). The third *k*-mer based tool, the RDP classifier ('dada2_rdp'), had significant problems even in these simple cases. The alignment-based VSEARCH tool ('qiime2_vsearch') showed low numbers of assigned species in both case 1 and 2 but results improved for higher taxonomic ranks. When the reference database was filtered to remove the sequences corresponding to either the species (case 3), genus (case 4), or family (case 5) of the query, SINTAX was more conservative than SKLEARN and made considerably fewer (incorrect) predictions at the ranks where reference sequences were missing. Interestingly, VSEARCH had the highest number of correctly assigned queries at the family level in case 3 (species missing) and case 4 (genus missing), indicating that this method works better than the *k*-mer based classifiers for higher taxonomic ranks when the reference database is incomplete.

Due to its overall good performance and robustness to parameter choice, SINTAX was chosen as the default option for taxonomic assignment in HAPP. However, it had a low classification rate at the order level when the family was missing in the database, which could be problematic for ecosystems with poor representation in the reference database. This was indeed observed when applying SINTAX on IBA data from Madagascar, but not when applied to the Swedish data (Fig 2b), consistent with a more extensive and less cataloged arthropod diversity in Madagascar. Considering that phylogenetic approaches are potentially more efficient when close reference sequences are lacking, we included such a method (EPA-NG) in the benchmarking, utilizing a phylogenetic tree of 49,358 insects derived by extension and quality filtering of a published tree [36] (see Methods). Indeed, annotation rate was doubled compared to SINTAX and approached 100% at

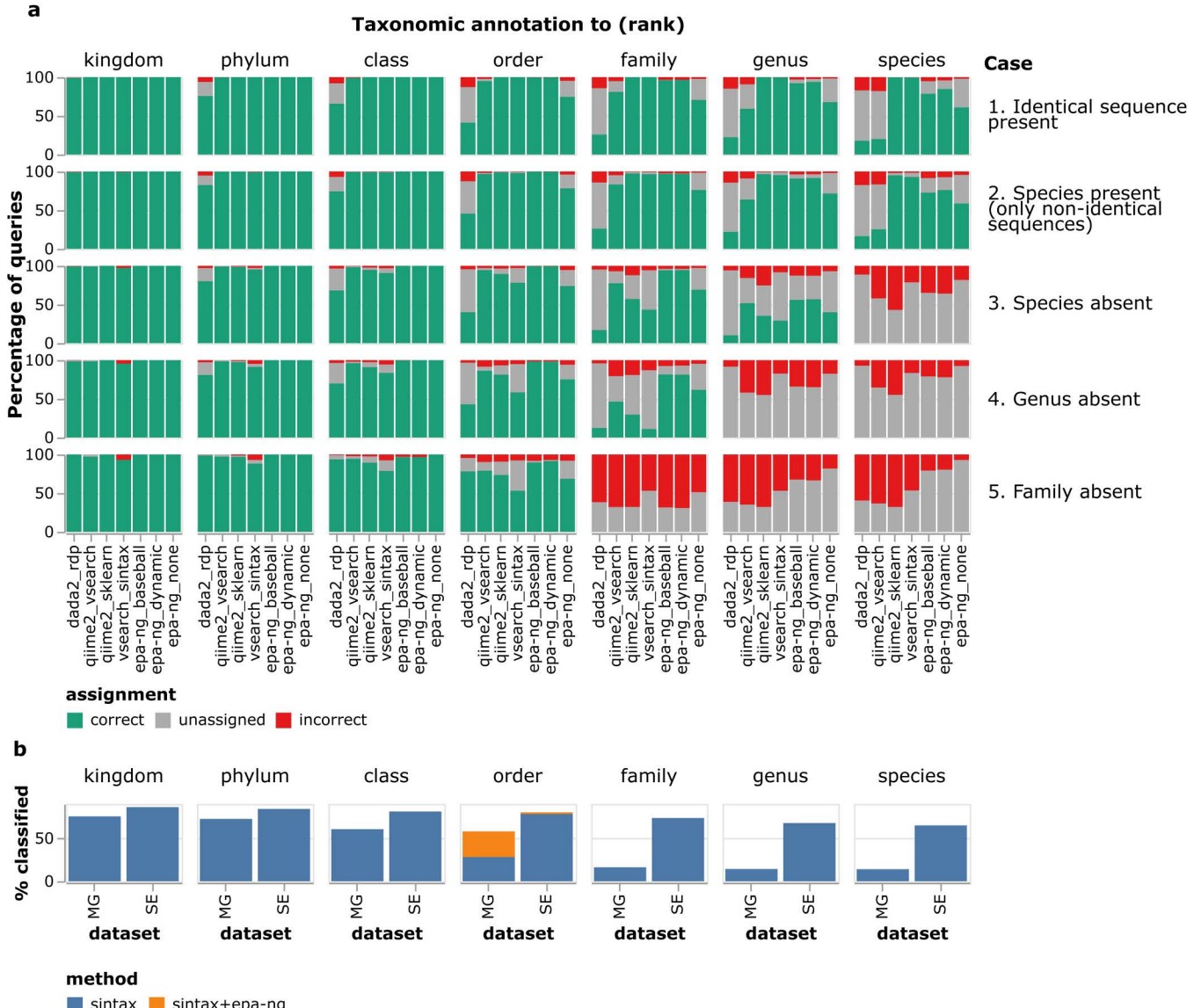

**Fig 2. Benchmarking of taxonomic assignment tools.** a. Barplots of correct, unassigned and incorrect sequences for different ranks and test cases. 'dada2_rdp' = DADA2 assignTaxonomy (an RDP classifier), 'qiime2_vsearch' and 'qiime2_sklearn' = QIIME2 feature-classifier vsearch and sklearn, respectively, 'vsearch_sintax' = SINTAX algorithm implemented in VSEARCH, 'epa-ng.baseball', 'epa-ng.dynamic' and 'epa-ng.none' = EPA-NG phylogenetic placement with baseball heuristics, dynamic heuristics and no heuristics, respectively, followed by assignments with GAPPA. b. Percentages of sequences classified at each rank on two real-world datasets from Madagascar ('MG', 701,769 ASVs) and Sweden ('SE', 821,559 ASVs) using the SINTAX algorithm. For the taxonomic rank 'order' we also show the percentage of classified sequences obtained when running EPA-NG (dynamic heuristics) on sequences that were annotated as unassigned Insecta by SINTAX, then taking order-level assignments from the EPA-NG result.

order level when the family was missing in the reference, and likewise drastically increased at family level when the genus was missing, while still giving low false annotation rates (Fig 2a). Applying it to the IBA ASVs annotated as class Insecta or Collembola by SINTAX but unclassified at the order level more than doubled the annotation rate at the order level for the Madagascar data (from 28 to 58%), but only marginally increased the rate for the Swedish data (78 to 80%), consistent with the *in silico* test (Fig 2b).

## Optimized sequence clustering to recover species-level OTUs

We benchmarked different ASV clustering methods on the Swedish IBA data, after chimeras had been removed, with performance being evaluated using the SINTAX assignments of ASVs to species. The precision (tendency of OTUs to consist of a single species) was high for all tools (in the interval 0.89-0.99; Fig 3a) but recall was lower, with mean values ranging from 0.09 to 0.82. This indicates that, under the parameters tested in our benchmark, the tools were prone to oversplit ASVs, *i.e.*, placing apparently conspecific ASVs into different clusters. This was especially evident for dbOTU3 which had the lowest recall value (0.025) of the tools tested.

Both SWARM and OptiClust use thresholds of sequence similarity to cluster ASVs, either in the form of maximum allowed 'differences' between sequences (SWARM, default = 1) or a maximum 'distance cutoff' (OptiClust, default = 0.03). The default settings resulted in low recall values (0.80 for OptiClust and 0.31 for SWARM). Increasing the maximum allowed differences for SWARM and the distance cutoff for OptiClust notably improved recall, which reached ≥0.9 for SWARM at differences ≥9 and for OptiClust at distance cutoff ≥0.04, with only marginal decreases in precision. However, for OptiClust, cutoff values >0.07 resulted in a sharp decrease in precision with little gain in recall. Assessing the performance using the F-score, which is the harmonic mean of precision and recall, we found that OptiClust with a cutoff of 0.05

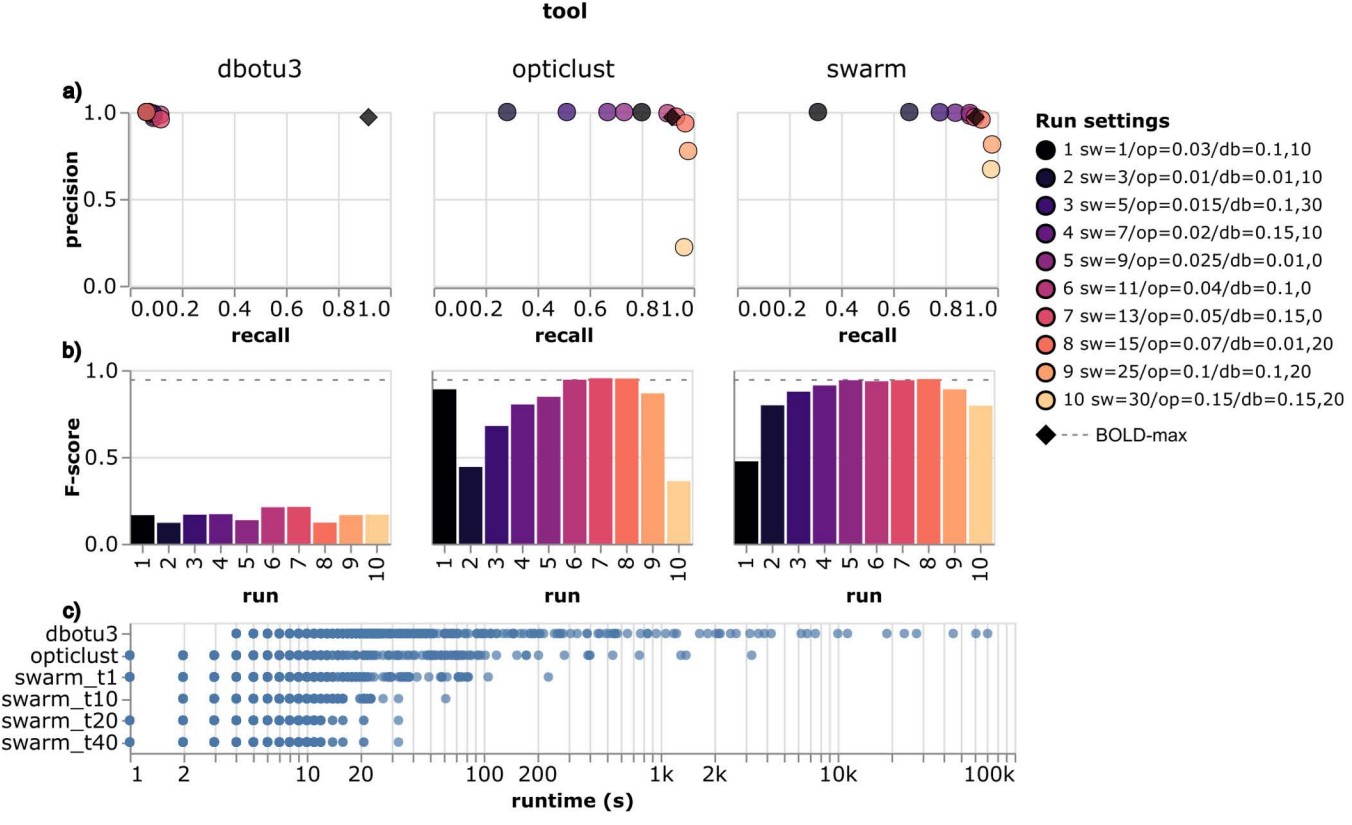

**Fig 3. Benchmarking of ASV clustering tools.** a. Precision vs. recall for each run of the different tools. Colors correspond to the different runs and the legend shows the settings used: 'sw' is the difference cutoff in Swarm, 'op' the distance cutoff in OptiClust, and 'db' the distance and abundance cutoffs in dbOTU3. The diamond symbol shows the precision and recall of the RESL-generated BINs in the BOLD database ('BOLD-max'). b. F-score values (harmonic mean of precision and recall) of the different runs for each tool. The dashed line shows the F-score of the BINs in the BOLD database. c. Runtime statistics for the different tools obtained from the same run settings. Each point shows the time to cluster ASVs belonging to a single taxonomic family. For SWARM, the number of threads is indicated on the y-axis ('swarm_t10' = 10 threads).

and SWARM with a difference of 15 gave the best, and highly similar, metrics (F-scores of 0.95 for both OptiClust and SWARM; Fig 3b).

To assess the performance of the open-source tools against the closed-source RESL algorithm [27], we downloaded the complete BOLD database (from 06-Jul-2022) and calculated precision and recall values (treating BIN assignments as 'ASV clusters') for 3,953,373 sequences with both BIN and species assignments (Fig 3a, diamond; Fig 3b, dotted line). The results show that RESL clustering of BOLD sequences into BINs performs on par with SWARM and OptiClust clustering of deep metabarcoding reads. Using closed-reference clustering of metabarcoding reads against BOLD gave similar results (S1 Fig).

In terms of runtime, SWARM and OptiClust showed similar performance in our benchmark, with the former slightly ahead; both tools were considerably faster than dbOTU3 (Fig 3c). Unlike the other tools, SWARM does not require a pairwise distance matrix to be computed beforehand, and it also has better multi-threading support than OptiClust. SWARM and OptiClust are still actively maintained, while dbOTU3 has been archived and has not been updated in the past 6 years. Because of its active developer community, its robust performance across settings, and its computational efficiency, we chose SWARM as the default clustering method in HAPP.

### A new algorithm for effectively removing NUMTs and other noise

Despite the extra chimera filtering step, we noted that the IBA data included a significant amount of noise, which was particularly obvious for well known groups in the Swedish fauna like Lepidoptera (butterflies and moths). However, existing noise filtering algorithms use overly simple criteria, focus on a single criterion, or are too computationally complex to be applied to deep metabarcoding data. Therefore, we designed a new, scalable algorithm based on a combination of criteria, NEEAT.

To address scalability, we apply NEEAT to clustered data rather than single ASVs, which has often been the case in previous algorithms. Also, we apply NEEAT to taxonomically partitioned datasets, allowing it to be run in parallel using the same strategy adopted by HAPP for other steps in the pipeline.

**Taxonomy**. In a detailed analysis of Swedish IBA data for Lepidoptera [7], we noted that taxonomic annotation failures at higher ranks are often associated with non-authentic CO1 sequences. Therefore, NEEAT includes a filter that removes clusters with annotation failures at higher taxonomic ranks. The rank at which failures are considered to signal noise is a user-provided setting (default order level). NEEAT uses three additional criteria to filter out noise clusters: echoes, evolutionary signatures, and read abundance.

**Echoes**. Like LULU, NEEAT searches across samples for unusual co-occurrence patterns among genetically similar clusters. Noise OTUs—both NUMTs and sequencing errors—may be expected to consistently occur in samples together with their parent OTUs (containing the corresponding authentic sequences) but in lower read numbers. Tuning parameters of this filter include the maximum distance between echo and parent cluster, the minimum fraction of echo samples with parent present, the max read ratio threshold and type ('max' or 'mean' across samples), and whether a significant correlation between echo and parent read numbers is required.

**Evolutionary signatures**. For protein-coding sequences, NEEAT compares evolutionary and edit distances among sequences to find clusters likely to represent NUMTs, off-target sequences, or chimeras missed in previous filtering steps. Specifically, we use an evolutionary distance taking chemical similarities between amino acids into account, and a metric that is sensitive to unexpectedly large evolutionary distances at small edit distances (see Methods). To improve performance, searches are limited to the $N$ nearest neighbors (in terms of edit distances), with $N$ being a tuning parameter. NEEAT uses two versions of this filter, a local filter requiring a minimum sample overlap between the clusters, and a global filter without this requirement.

**Abundance threshold**. Finally, NEEAT also implements an abundance threshold for the minimum number of reads for a cluster, based on either the mean or max across samples (default 'max').

We benchmarked NEEAT against abundance filtering, as in metaMATE, profile HMM filtering, and LULU. The tests used the Hexapoda (classes Insecta and Collembola) subset of the final Swedish IBA dataset, after chimera filtering, annotated by SINTAX and clustered by SWARM (d = 15) as described above. We measured false positives (authentic OTUs removed) as a weighted average of the number of unique BOLD BINs and trusted ASVs (see Methods) removed, and false negatives as a weighted average of the number of remaining duplicated BOLD BINs and spurious OTUs—OTUs in well-known insect families (>99% of expected species recorded [34]) not matching species in BOLD.

NEEAT significantly outperformed abundance filtering, HMM filtering and LULU abundance filtering in this benchmark (Fig 4a). In tests of individual filters, the NEEAT echo algorithm did slightly better than LULU (S2 Fig), and the NEEAT evolutionary algorithm (both local and global) outperformed HMM filtering by a large margin (S3 Fig). The performance of the abundance filter was insensitive to the type of reads used (raw, calibrated, proportional to sample reads, or proportional to non-spike-in reads); however, filtering on the max read number was better than filtering on the mean (S4 Fig). The false positive rate increased fairly rapidly with increasing thresholds for all major orders except the Diptera.

To assess the effectiveness of the taxonomic annotation filter, we compared the proportion of reads that would be removed by applying the criterion at different ranks for the IBA data from Sweden and Madagascar. The barcode reference library and reference tree include virtually all Swedish Hexapoda families, suggesting that most of the annotation failures at this level or above are likely to represent noise. As the noise level should be similar, a significant drop in the proportion of retained reads for Madagascar—with incomplete reference library and tree—would suggest that annotation failures include many false positives, that is, authentic OTUs. The results (Fig 4b) show that the phylogenetic placement method (EPA-NG) is considerably more robust to gaps in the reference tree than alignment-based (VSEARCH) or *k*-mer-based methods (SINTAX) are to gaps in the barcode library, despite the latter including two orders of magnitude more sequences. The drop is larger for reads than for unique ASVs or clusters, suggesting that annotation failures mainly affect the dominant taxa in Madagascar samples (S5 Fig). The lower annotation success of EPA-NG at the family level and below is likely due to poor coverage of the most species-rich insect families in the reference tree (Fig 4b).

Overall, the NEEAT algorithm is quite effective in reducing noise in the Swedish IBA data (Fig 4c and 4d). Detailed analysis of the Lepidoptera suggests that the data include roughly half the Swedish fauna of most families [7]. After NEEAT filtering, only a handful of well-known hexapod families fall significantly outside this range (Fig 4d).

### The HAPP pipeline provides a flexible and performant workflow

We designed HAPP as an open-source Snakemake [37] workflow that performs pre-filtering (ASV length and stop codons), chimera removal, taxonomic annotation, ASV clustering of non-chimeric sequences, and noise removal by NEEAT (Fig 5). The default choice of tools and settings reflects the benchmarking results described above but HAPP supports the full range of clustering and annotation tools included in the benchmarking, and can easily be extended to accommodate additional tools. The workflow supports splitting the input ASVs by taxonomy (e.g., by class or order) and running the clustering software on each split separately, with the assumption that ASVs assigned to different taxa at this level should not cluster together. This can be useful for large datasets because it lowers the number of pairwise comparisons and allows for parallelization of several steps in the workflow. HAPP is available at https://github.com/insect-biome-atlas/happ.

### Discussion

Deep metabarcoding projects come with unique challenges and opportunities. The deep sequencing and the large number of samples can reveal species that would otherwise have been missed. At the same time, the data comes with significant numbers of chimeras, NUMTs and other types of noise. Thus, utilizing the full potential of deep metabarcoding requires powerful bioinformatic tools.

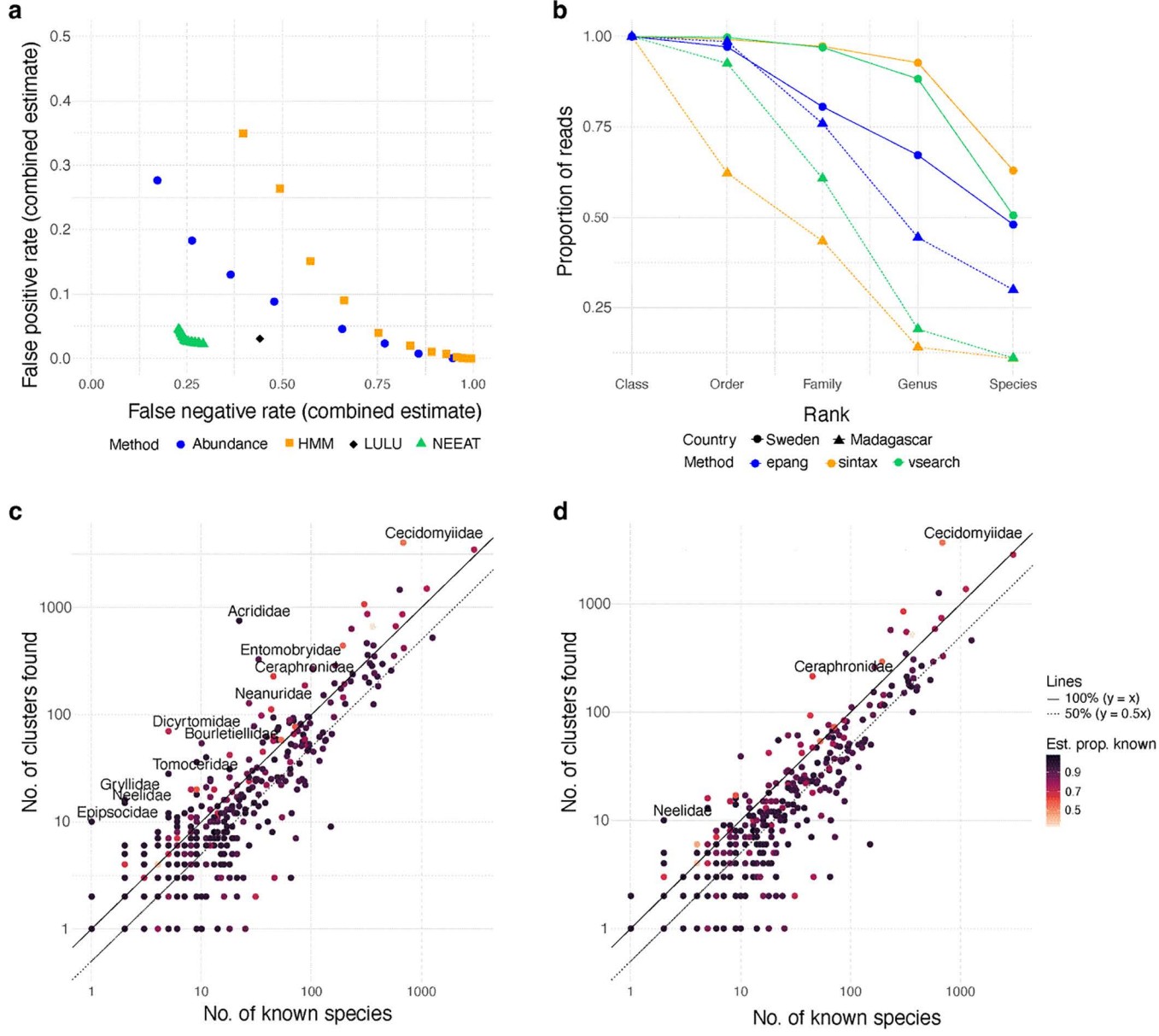

**Fig 4. Benchmarking of noise filtering tools.** a. Performance of noise filtering on the Swedish IBA data on hexapods using an abundance threshold (raw counts, with varying cutoff in terms of max number of reads), an HMM profile model (with varying bitscore cutoff), the LULU algorithm (default settings) and the NEEAT algorithm (Pareto front of settings explored here). b. Annotation success of different taxonomic annotation algorithms (sintax, a k-mer-based tool; vsearch, an alignment-based tool; and epang, a phylogenetic placement tool) on Swedish and Malagasy IBA data. c-d. The number of OTUs found in the Swedish IBA data before (c) and after (d) noise filtering with NEEAT, plotted against the number of species recorded from Sweden. Each point represents one family; the colors represent the estimated proportion of the total Swedish species diversity that is currently known [34]. Analysis of the Lepidoptera [7] suggests that the IBA data typically contains slightly more than 50% (dotted line) of the known species (full line) in well-known insect groups (dark purple). Poorly known families (lighter colors) may be represented by more species than currently known.

In this paper, we presented HAPP, a pipeline designed specifically for processing deep metabarcoding data. The pipeline includes optimised procedures for chimera removal, taxonomic annotation, OTU clustering and removal of noise OTUs. For the former steps, HAPP is based on existing tools, while a novel algorithm (NEEAT) was developed for the

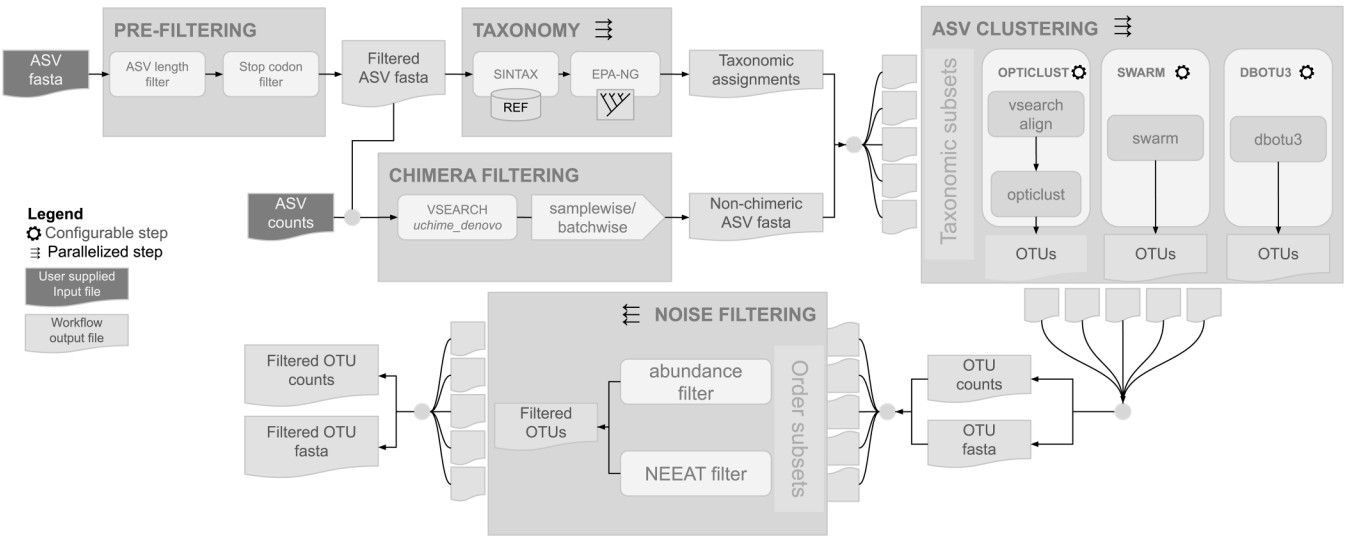

**Fig 5. Design of the HAPP pipeline.** After prefiltering to remove sequences of unexpected length or with in-frame stop codons, the pipeline involves four steps: (i) chimera filtering, (ii) taxonomic annotation, (iii) clustering and (iv) noise filtering. Parallelization is implemented by splitting the input into equally sized chunks in the taxonomic assignment step, and into taxonomic groups in the clustering and noise filtering steps. The image was generated with Google Drawings. The gear icon was obtained from https://openclipart.org/detail/202774/gear.

removal of noise OTUs. Each of the steps was optimized using a combination of *in silico* tests and benchmarking with real metabarcoding data.

The IBA data [6] used for benchmarking HAPP allowed us to perform tests across a range of settings, from small or moderately diverse groups with virtually complete reference libraries (many Swedish insects) to megadiverse taxa that are largely missing in current databases (some Swedish arthropods, most Malagasy taxa). Taken together, the benchmark results suggest that HAPP will be useful for many types of deep metabarcoding projects.

Our results emphasize the importance of filtering out chimeras from deep metabarcoding data. We found that the standard chimera filtering ('removeBimeraDenovo') in a commonly used ASV denoising pipeline (DADA2 [20]) was inadequate. Applying a second filtering pass in HAPP identified an additional 15% of ASVs in the Swedish IBA data as chimeras. Tweaking the DADA2 settings might bring down the frequency of chimeras, but the extra chimera removal in HAPP has a negligible rate of false positives, so there is little gain in bypassing it.

Among the taxonomic annotation tools we tested, SINTAX [22] performed best in our *in silico* tests using CO1 sequences from BOLD, in particular at lower taxonomic ranks. However, both our *in silico* tests and IBA data benchmarks revealed that SINTAX was overly conservative in assigning queries at higher taxonomic ranks. This was particularly obvious for the Madagascar data (Figs 4b and S5) despite new taxa at this level being unlikely to occur in the data. This profile contrasts starkly with that of phylogenetic placement using EPA-NG and Gappa [19,25,26], which performed well at higher taxonomic ranks while being unreliable at lower ranks (undoubtedly due to the poor coverage in the reference tree). This suggests that combining SINTAX and EPA-NG may be advantageous. We found that EPA-NG doubled the annotation rate at the order level for the Madagascar IBA data, while having only marginal effect on the Swedish data, consistent with the expected differences in the completeness of the reference libraries. We also noted that the success rate of annotating ASVs remained substantially higher for Sweden than for Madagascar, suggesting that there is room for further improvement (Fig 2b and S1 Table). As the available reference trees increase in coverage, we expect phylogenetic methods to gain ground. Given the current state, where no method can handle all contexts, we designed HAPP to support a range of taxonomic annotation tools and combinations of them.

BOLD and similar reference databases are invaluable resources for taxonomic annotation. However, relying on closed-reference clustering, where metabarcoding reads are assigned to clusters by matching them to existing records, is not adequate for the megadiverse and poorly covered organism groups often targeted in metabarcoding. Deep metabarcoding is likely to yield substantially more data per target OTU than contained in the reference databases, allowing OTUs to be circumscribed more effectively through de novo clustering. We think this explains why OptiClust [31] and SWARM [30] perform so well in our benchmarks, better than closed-reference clustering using BOLD BINs (S1 Fig). Clearly, the computational efficiency of OptiClust and SWARM is sufficient for de novo clustering of deep metabarcoding data. Unlike the RESL algorithm used by BOLD [27], both of these algorithms are open source, supporting reproducible science and community-based refinement.

Since HAPP clusters the data into OTUs, one might argue that first denoising the raw sequence data into ASVs is not necessary. However, there are several merits of inferring ASVs. ASVs make it faster and easier to reanalyse data at a later state - for example reclustering into OTUs, updating taxonomic annotations, and merging ASV tables from different studies - compared to starting with the raw data. Moreover, ASVs facilitate analyses of intra-specific diversity.

Filtering out data based on read numbers is often used as the main or sole cleaning step for metabarcoding data [14,38]. However, our results suggest that such abundance filtering is ineffective because deep metabarcoding data contain many authentic OTUs with few reads: the false positives rise quickly as the abundance threshold is increased (S4 Fig). However, because most groups seem to be associated with many low-read but authentic OTUs in our tests, we find it difficult to justify the use of taxon-specific thresholds as recommended by Andújar et al. (2021) [14].

The widely used LULU tool [13] can be understood as a special case of our echo algorithm, but we succeeded in improving performance slightly by ordering clusters first in terms of sample numbers and then read numbers, instead of just read numbers, allowing more echoes to be correctly identified when passing down the list. We also found that relaxing the read ratio threshold, targeting the mean rather than the max across samples, usually leads to performance gains, in contrast with LULU recommendations.

The work on profile HMMs [12] represents one of the most sophisticated attempts yet to use evolutionary signatures to separate noise from signal in metabarcoding data. However, our benchmarking indicates that it is quite ineffective compared to our approach, using pairwise comparisons of the ratio between edit distance and evolutionary distance. We think the reason for this is that the CO1 protein varies across the insect phylogeny, such that a single HMM profile is insufficient to accurately capture small deviations. This problem may be aggravated by taxon biases in the training data, resulting in poor fit of the model to the dominant groups in the metabarcoding data.

The optimal NEEAT performance was obtained with stringent settings for each of the component filters. The Pareto front was dominated by combinations using low thresholds for the read ratio in the echo filter (much lower than LULU defaults), high thresholds for the distance ratio in the evolutionary signature filter, and a low minimum max read number in the abundance filter. Our results suggest that annotation failures can also be used for noise filtering but that the taxonomic rank at which it is applied needs to be adjusted according to the completeness of the reference database. For instance, detailed analysis of the Swedish IBA data for Lepidoptera suggests that SINTAX annotation failures at the family level are often associated with noise ASVs in this dataset [7].

NEEAT is designed for deep metabarcoding data and may be less effective for smaller datasets. For instance, NEEAT separates NUMTs and sequencing errors from biological sequence variants by looking for the distribution of read ratios across samples. Authentic sequence variants are likely to be associated with significant spatial or temporal variation in read ratios, unlike NUMTs and sequencing errors. The more samples, the easier to separate these patterns. Similarly, a more complete sample of the fauna or flora facilitates the identification of NUMTs and sequencing errors through the deviations in pairwise evolutionary signatures used by NEEAT.

Even though HAPP makes substantial improvements over the state of the art, there is clearly room for further improvement. Both the echo and evolutionary signature algorithms are heuristic in that they start from the most abundant cluster,

assume it represents an authentic marker sequence, and then sequentially test and add sequences to the trusted set. An obvious improvement would be to revisit and continuously update the circumscription of the trusted set, even though it would increase the computational complexity. The success of the phylogenetic approach in improving the taxonomic annotation at higher ranks—despite the limited size of the reference tree—suggests that noise filtering could also be improved by considering phylogenetic information. The fact that our pairwise evolutionary signature algorithm outperformed the global HMM profile approach points in the same direction. Finally, switching to a proper probabilistic framework, and explicitly considering error models in that context, would also seem worthwhile exploring.

Despite the potential for further improvement, we think our benchmarks show that HAPP is sufficiently mature to be trusted for standard monitoring of well-known faunas, while at the same time being powerful enough to be used in exploring the diversity and composition of species groups or regional faunas that are largely unknown or poorly represented in reference databases.

## Methods

### Benchmarking datasets

**Deep metabarcoding data**. The deep metabarcoding data we used for benchmarking originate from the Insect Biome Atlas (IBA) project [6]. The project sampled the terrestrial arthropod faunas of Sweden and Madagascar using Malaise traps, soil samples and litter samples. The samples were metabarcoded using several different protocols, targeting a 418 bp region of CO1, were sequenced on the NovaSeq6000 platform. The raw sequences were preprocessed as described in Miraldo et al. (2024). Briefly, read trimming and filtering was applied using a Snakemake workflow available at https:// github.com/insect-biome-atlas/amplicon-multi-cutadapt, followed by filtering to remove sequences containing stop codons in the expected read-frame. Finally, the reads were denoised using the nf-core/ampliseq Nextflow workflow (v2.4.0) [39], which uses the DADA2 algorithm [20] to infer amplicon sequence variants (ASVs) from the preprocessed reads. This workflow also uses the DADA2 removeBimeraDenovo function for chimera removal using the following settings: 'method="consensus", minSampleFraction = 0.9, ignoreNNegatives = 1, minFoldParentOverAbundance = 2, minParentAbundance = 8, allowOneOff = FALSE, minOneOffParentDistance = 4, maxShift = 16'. We used two versions of the data for benchmarking: (1) an early version, restricted to mild lysis of the Malaise trap samples (4,560 samples from Sweden and 1,923 from Madagascar), containing 636,297 and 559,023 ASVs, respectively; and (2) a final version, containing CO1 data from all sample types (mild lysis and homogenates of Malaise trap samples, as well as soil and litter samples) (total of 5,804 and 2,111 samples containing 821,559 and 701,769 ASVs, for Sweden and Madagascar respectively). The data are available from https://doi.org/10.17044/scilifelab.25480681.v1 and https://doi.org/10.17044/scilifelab.25480681.v6.

**Taxonomic reference databases based on BOLD**. We generated a reference database for benchmarking taxonomic assignment tools from sequences in the BOLD database [27]. Firstly, sequence and taxonomic information for records in BOLD were downloaded from the GBIF Hosted Datasets (https://hosted-datasets.gbif.org/ibol/). This data was then filtered to only keep records annotated as 'COI-5P' and assigned to a BOLD BIN ID. The taxonomic information, as processed by GBIF, was parsed in order to assign species names and resolve higher level ranks for each BIN ID. Sequences were processed to remove gap characters and leading and trailing Ns. After this, any sequences with remaining non-standard characters were removed. Sequences were then clustered at 100% identity using VSEARCH [23]. This clustering was done separately for sequences assigned to each BIN ID. The processed database is available at the SciLifeLab Figshare repository [40]; we used version 4 of the database (https://doi.org/10.17044/scilifelab.20514192.v4). For the purposes of benchmarking, sequences were trimmed to the 418 bp region matching the barcode region used in this study. We refer to this in the following as the 'custom BOLD database'.

The above database adopts GBIF's name resolution algorithm for BINs with several taxonomic name annotations, using an 80% threshold for accepting a name as the consensus annotation. For gauging the performance of clustering tools against the RESL algorithm used to generate BOLD BINs, we also downloaded the complete BOLD database (from

06-Jul-2022). From this database, information on 3,953,373 sequences with both BOLD BIN and species assignments were extracted and used to compute precision and recall values, treating BIN assignments (the output of RESL) as 'ASV clusters'. No cleaning of taxonomic annotations was applied in this case.

For some benchmarks, we also used a carefully curated and almost complete barcode reference library for the Finnish fauna of arthropods [35], which is a small subset of BOLD (FinBOL). We expected this library to contain less noise than BOLD in general, and good coverage of the authentic CO1 ASVs likely to be encountered in Swedish arthropods. We refer to IBA ASVs with identical matches over the full ASV sequence to FinBOL reference sequences as 'trusted ASVs'.

### Chimera removal

We explored postprocessing chimera removal strategies for deep metabarcoding data based on the UCHIME de novo algorithm [18]. Specifically, we applied UCHIME with default settings in either a 'batchwise' or 'samplewise' mode. In the 'batchwise' mode, chimera detection is run directly on the full ASV dataset while in 'samplewise' mode, the ASV dataset is split into sample-specific fasta files containing sequences with a count > 0 in the sample. This means that the candidate parents for each ASV are restricted to the sequences present in the same sample. In addition, we implemented cutoffs on either the fraction of samples in which chimeric ASVs had to be present with their parents (for 'batchwise') or the fraction of samples in which ASVs had to be identified as chimeric (for 'samplewise'), testing both 'lenient' and 'strict' settings (Table 1).

We benchmarked the performance of the chimera detection on the initial version of the Swedish IBA data (636,297 ASVs), using the following evaluation metrics (with corresponding labels in Fig 1 in parenthesis): (1) total number of ASVs removed ('ASVs removed'); (2) proportion of the removed ASVs that were abundant, that is, found in more than one sample ('% abundant ASVs removed'); (3) proportion of ASVs identical to sequences in FinBOL removed ('% trusted ASVs removed'); (4) total number of reads removed ('Million reads removed'); (5) Number of reads of ASVs identical to FinBOL removed ('Trusted reads removed'); (6) generated clusters to the number of unique species annotations in the cleaned data ('Cluster:species ratio'); (7) number of species split across multiple clusters in the cleaned data ('Multicluster species"); and (8) precision, (9) recall and (10) F-score (harmonic mean of precision and recall) for the clustering ('Precision', 'Recall' and 'F-score', respectively), assessed against the SINTAX assignment of ASVs to species (see below).

Precision and recall were calculated as

$$precision = \frac{TP}{TP + FP}, \; recall = \frac{TP}{TP + FN} \tag{1}$$

**Table 1. Chimera detection settings. Four different strategies were tested. We used either a batchwise or samplewise approach, and then enforced a lenient or strict requirement on the parent-chimera co-occurrence pattern.**

| Name[1] | Mode | Settings |
|---|---|---|
| batchwise.strict | batchwise | ASVs had to share samples with both parents in ≧ 50% of samples in which they were present |
| batchwise.lenient | batchwise | ASVs had to share samples with both parents in ≧ 1 sample |
| samplewise.strict | samplewise | ASVs had to be identified as chimeric in all samples where they were present in order to be removed |
| samplewise.lenient | samplewise | ASVs had to be identified as chimeric in ≧ 1 sample in order to be removed |

**Notes**: [1]Method name used in Fig 1.

where TP = true positives, FP = false positives and FN = false negatives. TP was calculated by counting the number of times two ASVs belonging to the same ASV cluster also belonged to the same species, while FN was calculated by counting the number of times two ASVs that belonged to the same species were placed into different ASV clusters. FP was calculated as

$$FP = NP - TP \tag{2}$$

where NP is the total number of within-cluster pairs, that is the total number of possible ASV pairs within each cluster, summed over all clusters. For the evaluation metrics based on clustering, we used OptiClust [31] as implemented in mothur v1.44.11 [41] with a cutoff of 0.025.

## Taxonomic assignment

We benchmarked the performance of four different tools for taxonomic assignment of ASVs against a reference database: SINTAX [22] implemented in VSEARCH [23], DADA2 assignTaxonomy [20] (an RDP-classifier type of tool), and SKLEARN and VSEARCH implemented in Qiime2 via the 'classify-sklearn' and 'classify-consensus-vsearch' pipelines in the 'feature-classifier' plugin [42]. We also tested a phylogenetic placement tool (EPA-NG combined with GAPPA for taxonomic annotation) with three different settings for the heuristic placement algorithm used (Table 2). The reference database was the custom BOLD database described above. For the phylogenetic placement, we used a phylogenetic tree of 49,331 Hexapoda sequences with representative outgroups from taxa likely to be present in the IBA data. As a core, we used the tree of [36]. From this tree, we removed all sequences with frame shifts or stop codons, as well as all outgroup sequences (they were sprinkled across the tree); this gave us the ingroup tree. We then assembled CO1 data for relevant outgroups, missing Hexapoda classes (Collembola, Protura and Diplura), and a couple of ingroup representatives from GenBank and Bellini et al. (2023) [43], preferably complete CO1 sequences when available. The DNA sequences were converted to amino-acid sequences and aligned using MAFFT [44], and then converted back to a nucleotide alignment using PAL2NAL [45]. Relationships among the taxa in this dataset were inferred using MrBayes 3.2.7a [46] and a constrained backbone tree representing well established relationships and highly supported groups in the analysis of Bellini et al. (2023) [43]. Finally, the pruned Chesters tree was grafted onto a tree sampled from the posterior estimate from this analysis, replacing the ingroup representatives. All CO1 sequences in the final tree were realigned using MAFFT

**Table 2. Taxonomic annotation tools and settings.** We tested four different tools for annotations based on matching to a reference database (SINTAX, DADA2, SKLEARN and VSEARCH), and one tool (EPA-NG) with three different heuristics for phylogenetic placement in a reference tree.

| Name[1] | Tool | Type | Version | Settings (if different from default) | Reference |
|---|---|---|---|---|---|
| sintax | SINTAX | kmer | 2.21.2 (vsearch) | –sintax_cutoff 0.8 –randseed 15 –threads 1 | Edgar (2016), Rognes et al. (2016) |
| dada2_rdp | DADA2 | kmer | 1.30.0 | minBoot = 80 | Callahan et al. (2016) |
| qiime_sklearn | SKLEARN | kmer | 2023.9 (QIIME2) | | Bolyen et al. (2019) |
| qiime_vsearch | VSEARCH | alignment | 2023.9 (QIIME2), v2.22.1 (vsearch) | | Rognes et al. (2016), Bolyen et al. (2019) |
| epa-ng_baseball | EPA-NG | phylogenetic | | --baseball-heur | Barbera et al. (2019); Czech et al. (2020) |
| epa-ng_dynamic | EPA-NG | phylogenetic | | | Barbera et al. (2019); Czech et al. (2020) |
| epa-ng_none | EPA-NG | phylogenetic | | --no-heur | Barbera et al. (2019); Czech et al. (2020) |

**Notes**:

[1]Method name used in Fig 2.

and PAL2NAL as described above, followed by removal of all sites present in less than 90% of taxa; this resulted in a final alignment of the same length as the original Chesters alignment. The reference tree and all data and scripts used in obtaining it are available from https://github.com/insect-biome-atlas/paper-bioinformatic-methods.

For benchmarking, we generated five combinations of test/train datasets using the custom BOLD CO1 reference database described above. For each combination, 1000 species were randomly sampled and one sequence was selected from each species and added to the test dataset. Specifically, the examined cases (matching the cases in Fig 2) were (from simple to more challenging):

1. Keep all sequences in the reference database or tree, including the test sequences.

2. Remove all sequences identical to sequences in the test dataset from the reference database, but keep at least one sequence for each test species. For phylogenetic placement algorithms, use Insecta sequences from the reference database for which the species is represented in the tree but the specific sequence is missing.

3. Remove all species in the test dataset from the reference database, but keep at least one sequence for each test genus. For phylogenetic placement algorithms, use Insecta sequences from the reference database for which the species is missing but the corresponding genus is present in the reference tree.

4. Remove all genera in the test dataset from the reference database, but keep at least one sequence for each test family. For phylogenetic placement algorithms, use Insecta sequences from the reference database for which the genus is missing but the corresponding family is present in the reference tree.

5. Remove all families in the test dataset from the reference database, but keep at least one sequence for each test order. For phylogenetic placement algorithms, use Insecta sequences from the reference database for which the family is missing but the corresponding order is present in the reference tree.

For case 5 (family absent) only 490 species (instead of 1000) could be randomly sampled from the phylogenetic reference and placed into the test dataset.

In terms of benchmark metrics, we looked at the number of correctly assigned, incorrectly assigned and unassigned sequences for each of the five test cases. To further elucidate the performance "in the wild", with various completeness of the reference databases and in the presence of noise among query sequences, we compared the performance of the selected tools in taxonomic annotation of the final IBA datasets for Sweden and Madagascar.

## ASV clustering

We benchmarked the performance of three open-source tools for clustering metabarcoding ASVs into OTUs: (1) SWARM v3.1.0 [30]; (2) OptiClust [31] as implemented in mothur v1.44.11 [41]; and (3) dbOTU3 v1.5.3 [33].

For benchmarking, we used the initial IBA dataset for Sweden after removal of chimeras using the strict samplewise method described above. We ran each tool with a range of parameters (Table 3), and evaluated results by calculating precision and recall values given the SINTAX assignment of ASVs to species using the custom BOLD database, as described above.

We also wanted to compare the performance of these open-source tools to the closed-source RESL algorithm used to compute OTU clusters (BINs) for the BOLD database [27]. There are two ways in which RESL clustering performance can be compared to open-source tools for clustering of metabarcoding data: (1) open-source tools can be applied to the BOLD database, for which the RESL results are known (the BIN clusters); or (2) RESL results for a metabarcoding dataset can be inferred by closed-reference matching to the BOLD database, and interpreting the assignments of BINs to ASVs as the likely result of RESL clustering. Each of these methods has its advantages and disadvantages: The first approach compares the performance on a dataset of mostly authentic full-length CO1 barcode

**Table 3. Settings used in benchmarking ASV clustering tools. In all cases, 'eval1' represents default settings, which are shown in full. For other runs, only settings that differ from default values are shown.**

| Name¹ | Swarm | Opticlust | dbOTU3 |
|---|---|---|---|
| eval1 (default) | d = 1, fastidious = True, boundary = 3 | cutoff = 0.03 | dist = 0.1, abund = 10 |
| eval2 | d = 3 | cutoff = 0.01 | dist = 0.01, abund = 10 |
| eval3 | d = 5 | cutoff = 0.015 | dist = 0.1, abund = 30 |
| eval4 | d = 7 | cutoff = 0.02 | dist = 0.15, abund = 10 |
| eval5 | d = 9 | cutoff = 0.025 | dist = 0.01, abund = 0 |
| eval6 | d = 11 | cutoff = 0.04 | dist = 0.1, abund = 0 |
| eval7 | d = 13 | cutoff = 0.05 | dist = 0.15, abund = 0 |
| eval8 | d = 15 | cutoff = 0.07 | dist = 0.01, abund = 20 |
| eval9 | d = 25 | cutoff = 0.1 | dist = 0.1, abund = 20 |
| eval10 | d = 30 | cutoff = 0.15 | dist = 0.15, abund = 20 |

**Notes**:

¹Method name used in Fig 3.

sequences, that is, slightly longer sequences than a deep metabarcoding dataset, but also a database with a sparse and uneven coverage of the diversity of most biomes studied by metabarcoding. Thus, the results may not be relevant for deep metabarcoding data. The second approach potentially yields results that are more appropriate for clustering of metabarcoding data, but the RESL results are only indirectly inferred in ways that can be beneficial for RESL (its clusters are based on longer sequences, helping it interpret the cluster structure of the OTUs) or problematic for RESL (it does not have access to the better coverage of the studied biome represented by the deep metabarcoding data). We opted for using the latter method here (S1 Fig).

### Design of the NEEAT algorithm

Each NEEAT filter was coded in a separate R script. The filters were designed as follows.

**Echo filter**. The echo filter takes the OTU table ('counts') and the pairwise distance values from vsearch ('matchlist') as input, in addition to parameter settings. First, the OTUs are ordered by number of samples (primary criterion) and number of reads (secondary criterion), from largest to smallest. The top OTU is regarded as authentic. Each of the following OTUs is then compared to the *n* closest previously identified authentic clusters ('n_closest', default 10) within the desired distance ('min_match', default 84% identity). For each potential parent cluster, the algorithm checks whether it co-occurs with the potential echo in the required proportion of samples ('min_overlap', default 0.95), and whether the read number criteria are fulfilled. If 'require_corr' is TRUE and the parent and echo co-occur in more than three samples, then a linear model is fit to the read numbers of parent and echo, and it is checked whether the regression is significant (*p* value smaller than 'max_p_val', default 0.05) and the coefficient is smaller than 'max_read_ratio' (default 1.0). In all other cases, it is simply checked whether the read ratio is smaller than 'max_read_ratio'. The read ratio type ('read_ratio_type') can be set to either 'max' or 'mean' of reads across samples where both parent and echo occur.

**Evolutionary signature filter ('Evo' filter)**. This filter compares edit distances (computed with vsearch) to evolutionary distances between sequence pairs. To compute the latter, we first aligned ASV sequences by translating them to amino-acid sequences using the R package APE [47], aligning them with MAFFT [44] using the '--auto' option, and converting them back to nucleotide alignments using PAL2NAL [45]. We then computed two different metrics, 'dadn' and 'wdadn'. The former is simply the ratio between nonsynonymous (amino-acid) and synonymous differences. The latter weighted the amino-acid difference according to a biochemical similarity metric [48].

The evo filter orders OTUs in the same way as the echo filter. The top OTU is regarded as authentic, and the other OTUs are compared to authentic neighbors using 'min_match' and 'n_closest' criteria, as for the echo filter. In the local version of the evo filter ('require_overlap', default 'TRUE'), a minimum sample overlap ('min_overlap', default 0.95) is also required. If the ratio between the desired type of evolutionary distance ('dist_type', default 'wdadn') and the edit distance is larger than the threshold ('dist_threshold', default 1.0), the OTU is flagged as non-authentic.

**Abundance filter**. This filter simply flags OTUs with read numbers below the desired 'cutoff' as noise. The cutoff type ('cutoff_type') can be set to 'sum', in which case the sum of reads across samples is compared to the threshold, or to 'max', in which case the max reads in any sample is used instead.

**Taxonomic annotation filter**. This filter simply removes OTUs with uncertain annotation at the desired taxonomic rank ('assignment_rank', default 'Order'), relying on HAPP annotation conventions (see below).

**NEEAT**. In NEEAT, we apply the filters in sequence, using the retained OTUs from each step as input to the next. NEEAT first applies the echo filter and then the evo filter in local mode, the evo filter in global mode, the abundance filter, and finally the taxonomic annotation filter.

## Benchmarking of noise filtering algorithms

To benchmark noise filtering algorithms, we used the Swedish IBA data for orders of Hexapoda. The taxonomic assignment of each OTU was based on the assignment of the representative ASV, that is, the ASV with the highest median read number across samples of all ASVs included in the OTU. To measure false positives (authentic OTUs filtered out), we used a weighted average of the number of unique BOLD BINs and the number of trusted OTUs (the OTUs containing trusted ASVs from FinBOL) removed. To measure false negatives (non-authentic OTUs remaining) we used a weighted average of the number of duplicated BOLD BINs and the number of 'spurious' OTUs remaining. An OTU was considered spurious if it belonged to a well-known family (>99% of species known according to Ronquist et al. 2020) but none of the included ASVs matched a species in the reference library.

We first benchmarked the component filters individually, exploring a range of settings and using LULU [13] and an HMM profile model [12] as references for the echo and evo filter, respectively (S1–S4 Figs). We then computed the Pareto front of the NEEAT filter by combining individual filters under parameter settings with a low level of false positives (as false positives add up in NEEAT), and then removing those combinations that were dominated in the two performance dimensions by other combinations.

These tests filtered out all OTUs with uncertain annotations at the order level. To assess the performance of the annotation filter, we compared the annotation success for a fauna that is well covered in the reference library and phylogeny (IBA data from Sweden) with one that is not (IBA data from Madagascar). The proportion of noise OTUs should be similar for both faunas; thus, differences in annotation rate should be due to the effect of coverage in reference databases.

## Design of the HAPP pipeline

HAPP was designed as a Snakemake workflow. The user supplies two input files to the pipeline: a FASTA file of ASVs and a tab-separated file with counts of each ASV in all samples. The pipeline first pre-filters the ASVs, removing sequences that fall outside a user defined length interval (default: 403–418 bp) as well as those with in-frame stop codons. Taxonomy is assigned to the filtered ASVs using a two-step approach combining SINTAX and EPA-NG. Chimera filtering is applied to the filtered ASVs using the uchime_denovo algorithm and a user configurable filtering criteria (default: samplewise). The taxonomic assignments are then used to split the filtered, non-chimeric sequences into taxonomic subsets at a user configurable rank (default: Family). Each subset is then clustered into OTUs in parallel using all clustering software configured by the user (default: SWARM). The OTUs for all subsets are gathered per clustering software and are then split into order-level subsets. Each of these subsets are then filtered for noise in parallel, taking the abundance of OTUs into account as well as applying the NEEAT algorithm to OTUs.

**Additional filtering steps in HAPP**. HAPP includes several additional filtering steps not discussed above. First, it allows filtering out of clusters occurring in more than a specified fraction of negative controls, and extra abundance filtering (if the NEEAT filtering is deemed insufficient). Both of these filters are applied per dataset. The user controls the filters by supplying a metadata file specifying which samples are blanks/negative controls as well as which samples belong to what datasets.

HAPP also allows the user to filter out spike-in clusters based on user-provided species annotations or BOLD BIN annotations. As an alternative, an occurrence threshold (clusters of specified higher taxa occurring in more than x% of samples are deemed as spike-in clusters) can be used to identify and remove spike-in clusters.

**Handling of ambiguous taxonomic annotations**. HAPP uses several approaches to handle and resolve ambiguities in taxonomic annotations. An ASV is labeled as 'unassigned' if the taxonomic annotation algorithm failed to assign the ASV to a taxon at this rank. If the ASV is assigned to a taxon named 'Taxon_name' at the next higher rank, it is labeled as 'unassigned.Taxon_name'. If the ASV is unassigned at the next higher rank, the annotation at that level is simply copied to the lower rank. Ambiguities may also arise from the reference database itself. For the reference database we used, the taxon annotation of the sequences in each BOLD BIN was resolved by GBIF, if all sequences did not have the same taxon annotation in BOLD. Specifically, GBIF used an 80% cut-off to call a taxon annotation; BOLD BINs with more conflict among sequence annotations were not resolved to a name at that rank. In the case ASVs match such unresolved annotations in the reference database, HAPP labels the ASVs with the lowest rank that was fully resolved, followed by an underscore and one 'X' for each lower rank that was unresolved. For instance, a BIN in the family Hepialidae unresolved at the genus and species levels would be leveled at the species level as 'Hepialidae_XX'.

Further ambiguities could result in the clustering of ASVs into OTUs. HAPP uses a method similar to the GBIF name resolution mechanism in obtaining a consensus assignment. First, HAPP takes all taxonomic assignments from cleaned ASVs for each cluster, and then iterates from the lowest (most resolved) taxonomic rank to the highest. At each rank, taxonomic assignments for all ASVs in the cluster are gathered and weighted by their total read counts. If the weighted assignment make up 80% or more of the assignments at that rank, that taxonomy is propagated to the ASV cluster, including parent rank assignments. If no taxonomic assignment is above the 80% threshold, the algorithm continues to the parent rank. Taxonomic assignments at any available child ranks are set to the consensus assignment and prefixed with 'unresolved.'.

To illustrate the taxonomic annotation resolution algorithm in HAPP, consider the example in Table 4 of an ASV cluster with five ASVs.

At rank = Species the percentage of assignments would be Species1 = 20% ((10 + 10)/100), Species2 = 60% ((50 + 10)/100), Species3 = 20% (20/100) meaning no assignment reached above the 80% threshold. Moving up to rank = Genus the values would be Genus1 = 20%, Genus2 = 80% which would assign Genus2 as the consensus. The ASV cluster would then be given the taxonomic assignment 'Family1, Genus2, unresolved.Genus2'.

HAPP also computes the 'representative' ASV of each cluster as the ASV with the highest mean read number across samples of all ASVs belonging to the cluster. As an alternative to the consensus algorithm described above, it is also possible to use the assignment of the representative ASV for the entire OTU.

**Table 4. Example ASV cluster with five ASVs.**

| ASV_ID | Family | Genus | Species | Reads |
|---|---|---|---|---|
| asv1 | Family1 | Genus1 | Species1 | 10 |
| asv2 | Family1 | Genus1 | Species1 | 10 |
| asv3 | Family1 | Genus2 | Species2 | 50 |
| asv4 | Family1 | Genus2 | Species2 | 10 |
| asv5 | Family1 | Genus2 | Species3 | 20 |

## Supporting information

**S1 Fig. Benchmarking of ASV clustering tools, using the original species-level annotation of the BOLD sequences (rather than the GBIF-resolved BIN-level annotations), and assessing the putative performance of the RESL algorithm on IBA data through closed reference clustering according to the SINTAX assignments to BOLD BINs ('BOLD-closed').** a. Precision vs. recall for each run of the different tools. Colors correspond to the different runs and the legend summarizes the settings used: 'sw' gives the difference cutoff in Swarm, 'op' the distance cutoff in OptiClust, and 'db' the distance and abundance cutoffs in dbOTU3. The diamond symbol shows the result obtained in closed reference clustering using BOLD BIN assignments. b. F-score values (harmonic mean of precision and recall) of the different runs for each tool. The dashed line shows the F-score for closed reference clustering using BOLD BIN assignments.
(PDF)

**S2 Fig. Performance of the echo algorithm of NEEAT.** The plots show the performance on various hexapod taxa in terms of our combined estimates of false positives (authentic CO1 sequences removed) and false negatives (false CO1 sequences remaining), and compares it to the performance of LULU. Four different settings were explored: 'Corr. max' = enforcing a significant correlation in read numbers between noise and signal, varying max read ratio from 0.05 to 1.0. 'Corr.mean' = ditto, varying mean read ratio from 0.05 to 1.0. 'No.corr.max' = no correlation test, varying max read ratio threshold from 0.1 to 1.0. 'No.corr.mean' = ditto, varying mean read ratio from 0.1 to 1.0. 'LULU' = LULU algorithm with default settings (similar to 'No.corr.max' with read ratio of 1.0).
(PDF)

**S3 Fig. Performance of the evolutionary signature algorithm of NEEAT.** The plots show the performance on various hexapod taxa in terms of our combined estimates of false positives (authentic CO1 sequences removed) and false negatives (false CO1 sequences remaining), and compares it to the performance of the HMM profile algorithm. Four different settings were explored: 'dAdN.global' = unweighted amino acid distances, comparing sequences across all samples, varying distance threshold from 1.0 to 6.0. 'dAdN.local' = ditto, comparing sequences with sample overlap, varying distance threshold from 0.4 to 2.7. 'dWAdN.global' = biochemically weighted amino acid distances, comparing sequences across all samples, varying distance threshold from 0.8 to 4.0. 'dWAdN.local' = ditto, comparing sequences with sample overlap, varying distance threshold from 0.1 to 1.7. 'HMM' = HMM profile algorithm, varying bitscore threshold from 160 to 300.
(PDF)

**S4 Fig. Performance of the abundance threshold algorithm of NEEAT.** The plots show the performance on various hexapod taxa in terms of our combined estimates of false positives (authentic CO1 sequences removed) and false negatives (false CO1 sequences remaining). We explored four different types of read counts: 'calibrated' = read counts calibrated by the number of biological spike in reads. 'raw' = raw read counts. 'sample_proportional' = the proportion of the reads in the sample, with the biological spike in reads excluded. 'tot_proportional' = the proportion of the reads in the sample, with biological spike in reads included. We explored two types of thresholds: 'max' = maximum reads across samples. 'sum' = the sum of reads across samples. The threshold was varied from 1 to 500; for proportional counts, the threshold was divided by 1E6 (the expected number of reads per sample).
(PDF)

**S5 Fig. Taxonomic annotation success of different algorithms for faunas with different coverage in reference databases.** The plots show the annotation success for IBA data at taxonomic ranks ranging from phylum to species for a k-mer based method ('sintax'), an alignment-based method ('vsearch'), and a phylogenetic placement method ('epang'). Results are shown for a fauna well covered in the BOLD reference database and the reference tree used for annotation

('Sweden') and for a poorly covered fauna ('Madagascar'). The results for clusters are based on the taxonomic annotation for the representative ASV of each cluster.
(PDF)

**S1 Table. The remaining number of sequence reads, ASVs, OTUs, annotated BOLD-BINS, and annotated species after different steps of HAPP running on the IBA data from Sweden and Madagascar.** Species and BOLD BIN numbers are provided both based on individual ASV annotations and based on consensus ('cons.') annotations of OTUs. OTU clustering is applied only after chimera removal in HAPP, which explains the NA values. 'After_cleaning' refers to the last (optional) cleaning step where OTUs corresponding to spike-in controls and OTUs occurring in more than a specified fraction of negative control samples are removed.
(PDF)

## Author contributions

**Conceptualization:** Anders F. Andersson, Fredrik Ronquist.

**Data curation:** Elzbieta Iwaszkiewicz-Eggebrecht, Laura J.A. van Dijk, Robert Goodsell, Nerivania N. Godeiro, Bruno C. Bellini, Johanna Orsholm, Piotr Łukasik, Andreia Miraldo.

**Formal analysis:** John Sundh, Emma Granqvist, Anders F. Andersson, Fredrik Ronquist.

**Funding acquisition:** Tomas Roslin, Ayco J.M. Tack, Anders F. Andersson, Fredrik Ronquist.

**Software:** John Sundh, Lokeshwaran Manoharan, Fredrik Ronquist.

**Supervision:** Anders F. Andersson, Fredrik Ronquist.

**Writing – original draft:** John Sundh, Emma Granqvist, Anders F. Andersson, Fredrik Ronquist.

**Writing – review & editing:** John Sundh, Emma Granqvist, Elzbieta Iwaszkiewicz-Eggebrecht, Lokeshwaran Manoharan, Laura J.A. van Dijk, Robert Goodsell, Nerivania N. Godeiro, Bruno C. Bellini, Johanna Orsholm, Piotr Łukasik, Andreia Miraldo, Tomas Roslin, Ayco J.M. Tack, Anders F. Andersson, Fredrik Ronquist.

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
