## [Decision Letter · Decision Letter 0]

11 Jun 2025

HAPP: High-Accuracy Pipeline for Processing deep metabarcoding data

PLOS Computational Biology

Dear Dr. Andersson,

Thank you for submitting your manuscript to PLOS Computational Biology. After careful consideration, we feel that it has merit but does not fully meet PLOS Computational Biology's publication criteria as it currently stands. Therefore, we invite you to submit a revised version of the manuscript that addresses the points raised during the review process.

Please submit your revised manuscript within 60 days Aug 11 2025 11:59PM. If you will need more time than this to complete your revisions, please reply to this message or contact the journal office at ploscompbiol@plos.org. Please include the following items when submitting your revised manuscript:

We look forward to receiving your revised manuscript.

Kind regards,

Youhua Chen

Academic Editor

PLOS Computational Biology

Tobias Bollenbach

Section Editor

PLOS Computational Biology

**Additional Editor Comments :**

You can see that both reviewers see the merits of the paper and therefore I encourage the authors to revise the paper according to the comments. Please note that the second reviewer also provided comments on the text and uploaded as a readable file, please read and revise as well.

**Journal Requirements:**

1) Your manuscript is missing the following sections: Design and Implementation, and Availability and Future Directions. Please ensure that your article adheres to the standard Software article layout and order of Abstract, Introduction, Design and Implementation, Results, and Availability and Future Directions. For details on what each section should contain, see our Software article guidelines:

https://journals.plos.org/ploscompbiol/s/submission-guidelines#loc-software-submissions

3) Some material included in your submission may be copyrighted. According to PLOSu2019s copyright policy, authors who use figures or other material (e.g., graphics, clipart, maps) from another author or copyright holder must demonstrate or obtain permission to publish this material under the Creative Commons Attribution 4.0 International (CC BY 4.0) License used by PLOS journals. Please closely review the details of PLOSu2019s copyright requirements here: PLOS Licenses and Copyright. If you need to request permissions from a copyright holder, you may use PLOS's Copyright Content Permission form.

Potential Copyright Issues:

i) Figure 5. Please confirm whether you drew the images / clip-art within the figure panels by hand. If you did not draw the images, please provide (a) a link to the source of the images or icons and their license / terms of use; or (b) written permission from the copyright holder to publish the images or icons under our CC BY 4.0 license. Alternatively, you may replace the images with open source alternatives. See these open source resources you may use to replace images / clip-art:

4) Please amend your detailed Financial Disclosure statement. This is published with the article. It must therefore be completed in full sentences and contain the exact wording you wish to be published.

3) If any authors received a salary from any of your funders, please state which authors and which funders.

5) Please ensure that the funders and grant numbers match between the Financial Disclosure field and the Funding Information tab in your submission form. Note that the funders must be provided in the same order in both places as well.

6) Please provide a completed 'Competing Interests' statement, including any COIs declared by your co-authors. If you have no competing interests to declare, please state "The authors have declared that no competing interests exist". 

**Reviewers' comments:**

Reviewer's Responses to Questions

**Comments to the Authors:**

**Please note that one of the reviews is uploaded as an attachment.**

Reviewer #1: PCOMPBIOL-D-25-00687

The study by Sundh et al. describes a pipeline for processing metabarcoding data, primarily focused on insect communities using coding sequences (specifically CO1). The pipeline, HAPP, integrates previously published bioinformatic tools and introduces a new algorithm, NEEAT, to improve denoising. The benchmarking results are informative to implement the best tools and parameters as well. However, I have several questions regarding the pipeline:

- What is the purpose of using DADA2 before clustering? Have the authors tested the pipeline without this step to evaluate whether DADA2 is truly necessary in this context?

- Figure 5 is confusing. The pipeline generates ASVs, optionally removes chimeras, assigns taxonomy, then clusters into OTUs and removes noise using NEEAT. Assigning taxonomy before clustering and denoising seems counterintuitive and potentially time-consuming. Could the authors clarify their rationale for this order of operations?

- I am also puzzled by the choice of SINTAX over VSEARCH, especially given that the authors mention VSEARCH performed better. In most community ecology studies, we lack complete knowledge of all species present in a community. Therefore, it would make more sense to prioritize tools that perform best when genus or species information is missing. Could the authors clarify this choice? Additionally, what is the proportion of incorrectly assigned ASVs (at the species, genus, and family levels) that were nonetheless assigned to the closest taxon in terms of phylogeny? In other words, how many ASVs were incorrectly assigned when the correct species was absent, but were assigned to the most closely related species available in the database?

- While the pipeline is interesting, it appears to have limited applicability. Is there a version of the pipeline that can be used with SSU rDNA, which is the most commonly used marker? Can it be used on another set of taxa for example eDNA from marine samples?

I have also a few questions about NEEAT:

- What happens if only a limited amount of data is available for reference, especially for the Evolutionary signature part?

- How do you make the difference between error and population variation?

- Why DADA2 followed by clustering do not eliminate the noise? What do you mean by recurring sequencing error?

Minor comments:

I recommend that the authors cite the original publications describing the tools listed in Table 2, along with their respective versions, rather than only referencing the version of QIIME2. For example, the authors should cite Rognes et al. (2016) for VSEARCH instead of citing the QIIME2 publication, and should use the latest version of the tool (v2.30). I was unable to determine which version of VSEARCH is implemented within QIIME2.

Some references are needed for example L 90-93.

Reviewer #2: The manuscript by Sundh and collaborators presents a pipeline (HAPP) for processing denoised amplicons from deep (>2M reads/sample) metabarcoding, including a new process, NEEAT, that eliminates spurious sequences.

The authors test their work with two datasets of insect samples metabarcoding, from Sweden and Magadascar, using a fragment of the COI gene. The pipeline is available on github.

The manuscript explains the particularities of deep metabarcoding , although it does not mention the effect of the binned Qscores offered by the Illumina platforms used for deep metabarcoding. The starting point of the HAPP is the ASV table from dada2, but the estimation of the error rates in which dada2 relies for its denoising is not informative when only 3 or 4 Qscores are given. But using DADA2 is, I think, irrelevant for this pipeline. The default setup for secondary clustering in HAPP (using swarm) is d = 12. That implies that sequences with up to 12 unseen haplotypes linking them will end up in the same OTU. It is very, very unlikely that DADA2 is doing anything in this regard: raw sequences will have way fewer errors corrected by DADA2 (this info can be extracted from dada2 ). in other words, I think doing dada2 + swarm with a high d offers the same output as swarm with a high d from the derep data. Particularly if the error object is not very useful.

For the purposes of my review, I will ask the authors to address the Qbinned scores and their effect in ASV estimation. I think it makes a great point in favor of using HAPP, given that the ASV estimation on its own might be misleading.

There are some other points attached as comments in the pdf of the manuscript.

With regards to the NEEAT algorithm, I think it poses a move in the right direction towards eliminating spurious sequences that may arise from sequencing. However, I haven't been able to explore its output, as it is embedded in HAPP and I havent managed a succesfull run of the pipeline. I think it would be useful to have a standalone version of NEEAT, so that it can be tested with other datasets and compared with other methods for spurious sequence removal.

I have installed the HAPP pipeline in two different systems: a windows laptop running WSL2 with Ubuntu 20.4 and a server running Ubuntu 20.4. In both cases, I modified the config yaml and run it with the example data provided. I would appreciate if the config/config.yaml file would be in such a way that the only thing to change would be the full path to the data. That will make it easier on the user to try the pipeline.

In my windows laptop, this was the error I got when running the pipeline:

Error in rule parse_qiime:

jobid: 3

input: results/taxonomy/vsearch/test/taxonomy.raw.tsv

output: results/taxonomy/vsearch/test/taxonomy.tsv

log: logs/parse_qiime/vsearch/test.log (check log file(s) for error details)

shell:

python /home/rgallego/.cache/snakemake/snakemake/source-cache/runtime-cache/tmp9imfwq69/file/home/rgallego/local_review/happ/workflow/rules/../scripts/parse_qiime.py results/taxonomy/vsearch/test/taxonomy.raw.tsv results/taxonomy/vsearch/test/taxonomy.tsv -r Kingdom Phylum Class Order Family Genus Species BOLD_bin > logs/parse_qiime/vsearch/test.log 2>&1

(one of the commands exited with non-zero exit code; note that snakemake uses bash strict mode!)

With full log file not adding much information.

In my server, I got the following error:

Error in rule filter_seqs:

jobid: 2

input: data/test/asv_counts.tsv, results/chimera/test/filtered/chimera1/samplewise.uchime_denovo/nonchimeras.fasta, /home/meg/rgallego/happ/data/test/asv_taxonomy.tsv

output: results/common/test/chimera1/samplewise.uchime_denovo/Family/taxa

log: logs/filter_seqs/test/chimera1/samplewise.uchime_denovo/Family.filter.log (check log file(s) for error details)

I understand that the troubleshooting of the pipeline is not the main focus of the manuscript, but it is hard to finish a review without a chance of seeing the output of the pipeline.

So I would mark this as major revision, not because there are substantial flaws in the manuscript, but because I would like to be able to explore the output of the pipeline. I would suggest to the authors to provide a standalone version of NEEAT, so that it can be tested with other datasets and compared with other methods for spurious sequence removal.

**Have the authors made all data and (if applicable) computational code underlying the findings in their manuscript fully available?**

Reviewer #1: Yes

Reviewer #2: Yes

PLOS authors have the option to publish the peer review history of their article (what does this mean? ). If published, this will include your full peer review and any attached files.

**Do you want your identity to be public for this peer review?** For information about this choice, including consent withdrawal, please see our Privacy Policy .

Reviewer #1: **Yes: ** Jean-David Grattepanche

Reviewer #2: **Yes: ** Ramon Gallego

**Figure resubmission:**

**Reproducibility:**



---

## [Decision Letter · Decision Letter 1]

24 Sep 2025

Dear Professor Andersson,

We are pleased to inform you that your manuscript 'HAPP: High-Accuracy Pipeline for Processing deep metabarcoding data' has been provisionally accepted for publication in PLOS Computational Biology.

Best regards,

Tobias Bollenbach

Section Editor

PLOS Computational Biology

Reviewer's Responses to Questions

**Comments to the Authors:**

Reviewer #1: The clarifications provided by the authors are helpful and improve the manuscript.

I have a final comment regarding the use of DADA2. While I understand the motivation of saving time and computational resources, what is the likelihood that DADA2 may have incorrectly assigned noise to real data—cases that your tool could have identified and assigned correctly? I recognize that variability within COI is different compared to SSU data, but it would be helpful to include some discussion of this point in the manuscript.

Reviewer #2: The authors have adressed my questions.

**Have the authors made all data and (if applicable) computational code underlying the findings in their manuscript fully available?**

Reviewer #1: Yes

Reviewer #2: Yes

PLOS authors have the option to publish the peer review history of their article (what does this mean? ). If published, this will include your full peer review and any attached files.

**Do you want your identity to be public for this peer review?** For information about this choice, including consent withdrawal, please see our Privacy Policy .

Reviewer #1: **Yes: ** Jean-David Grattepanche

Reviewer #2: **Yes: ** Ramon Gallego

---

## [Editor Report · Acceptance letter]

PCOMPBIOL-D-25-00687R1

HAPP: High-Accuracy Pipeline for Processing deep metabarcoding data

Dear Dr Andersson,

I am pleased to inform you that your manuscript has been formally accepted for publication in PLOS Computational Biology. Your manuscript is now with our production department and you will be notified of the publication date in due course.

With kind regards,

Anita Estes
